# Feeding Postures and Substrate Use of François’ Langurs (*Trachypithecus francoisi*) in the Limestone Forest of Southwest China

**DOI:** 10.3390/ani14040565

**Published:** 2024-02-08

**Authors:** Shiyi Lu, Nanxin Lin, Anshu Huang, Dewen Tong, Yongyan Liang, Youbang Li, Changhu Lu

**Affiliations:** 1College of Biology and the Environment, Nanjing Forestry University, Nanjing 210037, China; lushiyi.cool@163.com; 2Guangxi Forest Resources and Environment Monitoring Center, Nanning 530028, China; amice110@126.com (A.H.); tongdewen@163.com (D.T.); lyy913@163.com (Y.L.); 3Observation and Study Station of National Forest Ecosystem in Guangxi Dayaoshan, Laibin 546100, China; 4Guangxi Key Laboratory for Polysaccharide Materials and Modifications, Key Laboratory of Protection and Utilization of Marine Resources, School of Marine Sciences and Biotechnology, Guangxi Minzu University, Nanning 530008, China; nanxinlingxmzu@163.com; 5College of Life Sciences, Guangxi Normal University, Guilin 541006, China

**Keywords:** François’ langurs, feeding posture, forest strata height, substrate selection

## Abstract

**Simple Summary:**

Adaptations to environments are crucial for wild animals to survive changing habitats. We study the feeding postures and forest strata use of François’ langurs to explore how these langurs behaviorally adapt to limestone forests. We found that these langurs used sitting as the most frequent feeding posture and tended to use the lower and middle forest during feeding, which is likely linked to food resource availability. Our result demonstrates that the spatial distribution of foods in the limestone forest has an important effect on the feeding posture of the François’ langurs and their forest layer utilization, highlighting the importance of understanding the effect of ecological factors on the adaptation of the langurs to the unique limestone habitat.

**Abstract:**

The feeding posture of a group of François’ langurs in Fusui County, Guangxi, was studied using instantaneous scan sampling from January to December 2016 to explore how the species adapts to karst limestone forests by collecting data on feeding posture, forest strata height, and substrate use. The results showed that leaves were the main food type of the François’ langurs, with young leaves accounting for 64.97% ± 19.08% of the food composition, mature leaves accounting for 11.88% ± 12.09%, fruits accounting for 12.96% ± 12.89%, flowers accounting for 4.16% ± 4.06%, and other food types, including stems, petioles, and other unknown parts of the tree, accounting for a total of 6.03% ± 9.09%. The François’ langurs had four main postures during feeding, of which sitting and bipedal standing feeding accounted for the largest proportions, at 85.99% ± 5.97% and 12.33% ± 6.08% of the total records, respectively. Quadrupedal standing and suspending were rarely observed and only appeared occasionally during feeding activities at the peak resting period, the two postures together accounting for 1.39% ± 1.59% of the total records. The feeding postures of the langurs had marked seasonal variation, as evidenced by the fact that seated feeding accounted for a significantly higher proportion of the total behavioral records in the rainy season than in the dry season, whereas feeding while standing bipedally was significantly more frequent during the dry season. Correlation analyses showed that feeding posture was correlated with food composition, showing a positive correlation between the proportion of bipedal standing feeding and mature leaf consumption. François’ langurs preferred to forage in the lower and middle forest layers, with the lower forest layer accounting for 55.93% ± 16.50% of the total number of recordings and the middle forest layer accounting for 33.63% ± 18.33%. Langurs were less likely to forage on the ground (rocks), accounting for only 6.79% ± 4.78% of the records. The frequency of langurs feeding in the upper part of the forest layer was the lowest at 3.65% ± 2.73%. Additionally, in the dry season, langurs utilized the lower forest layer more but used the middle forest layer less than in the rainy season. This study demonstrates that the spatial distribution of foods in the limestone forest has an important effect on the feeding posture of François’ langurs and their forest layer utilization.

## 1. Introduction

Postural behavior reflects the relationship between the environment and an animal’s morphological structure and behavior [1]. Among mammals, primates have the most diverse postural behaviors [2] and are thus better able to adapt to complex environments [3]. Studying postural behavior in primates can provide important data for a deeper understanding of their morphological anatomy, ecology, and environmental adaptations [2,4]. Primate postural behavior is not only influenced by their morphological structure [5,6,7], physiological characteristics [8,9], and individual developmental characteristics [9,10] but also influenced by their habitat’s structure [1,11,12], spatial distribution patterns of food resources, and their seasonal variations [13,14,15], as well as temperature [16].

Feeding posture is an important component of an animal’s postural behavior. Due to its close relationship with food acquisition ability and efficiency, feeding posture characteristics directly reflect an animal’s survival potential and ability to adapt to environmental changes [2,3,17]. Among many ecological factors, the spatial distribution of food resources is one of the most important factors affecting the feeding posture of animals [18,19]. In nature, primates experience cyclical changes in the distribution pattern of food resources, manifesting as seasonal fluctuations in food quantity and quality [20]. Primates adopt postural adjustment strategies in response to seasonal changes in food resources [17]. For example, some primates forage more frequently in a bipedal standing position during the dry season when food is scarce and more frequently in a sitting position during the rainy season when food is abundant [13]. Due to differences in the distribution patterns of various foods, primates adopt specific postures to reach food based on the balance between feeding benefits and the risk of falling from the trees. For example, the red howler monkeys (*Alouatta seniculus*) in French Guiana’s primary rainforest heavily depend on leaves for their diet during the dry season and thus travel more frequently by quadrupedal walking on large supports because feeding on leaves by quadrupedal walking is probably energy-inexpensive and relatively stable [18]. During the wet season, howlers feed more frequently by sitting because the abundant fruits that they eat then require more time for special manipulation [18].

Other ecological factors also have an impact on the posture of animals, such as ambient temperature [16,21,22] and habitat structure [1]. Homeotherms reduce metabolic costs through thermoregulation [16] or by obtaining heat from the outside world to prevent excessive bodily energy consumption at extreme temperatures [23]. For example, during the cold season, Japanese macaques *(Macaca fuscata*) regulate their body surface temperature by sunbathing and sitting in groups to reduce heat loss [22,23]. Ring-tailed lemurs (*Lemur catta*) on the island of Madagascar adopt a huddled position to prevent heat loss [24]. In contrast, Colombian white-faced capuchins (*Cebus capucinus*) dissipate heat during the hot season by exposing their tongues and lying on their backs [25]. Feeding posture has an important effect on feeding benefits. Furthermore, differences in habitat structure may create differences in the postural behavior of langurs living in the same environment [1]. Habitat characteristics determine animals during feeding and microhabitat utilization [1,13,26]. For example, *Trachypithecus delacouri* extensively uses the rocky ground during movement in sparsely vegetated areas of Vietnam with few trees [1].

François’ langur *(Trachypithecus francoisi*) is an endemic primate living in karst mountains, belonging to the family Cercopithecidae and subfamily Colobinae, which mainly feeds on tree leaves, flowers, and fruits [27]. This species is exclusively found in Northern Vietnam and Guangxi, Guizhou, and Chongqing in China, and is the species of the genus *Trachypithecus* at the northern edge of its distribution area [28]. Studies on the postural behavior of François’ langurs have been reported. However, previous studies have focused on langurs at the Nonggang National Nature Reserve in Guangxi; nothing is known about the postural behavior of François’ langurs outside of this area [13]. The vegetation in Guangxi Nonggang National Nature Reserve is very well protected, while outside the reserve, the habitat of the François’ langurs has been severely fragmented, and the vegetation structure has been damaged for historical reasons because all the flat zones among langur habitats are legally used by residents. In addition, differences in habitat quality can lead to significant changes in the behavioral and ecological characteristics of animals, especially in the amount and spatial distribution of food resources. The feeding postures of François’ langurs have not been reported. In this study, we first reported the dietary composition and then described the feeding posture and forest strata used by François’ langurs during feeding. Finally, we examined the seasonal variations in feeding postures and substrate use.

## 2. Materials and Methods

### 2.1. Study Site and Subjects

The study site was located at Nonghedian (local name), Zhonghuacun, Changping Township, Fusui County, Guangxi, China (22°41′16″ N; 107°49′11″ E). This area consists of exposed karst landforms, dominated by peaked depressions and valleys, with peaks 300–700 m above sea level. The vegetation type is a seasonal rainforest in limestone mountains with predominately more tropical species, mostly tall trees, and abundant vines. In the plains of the valley bottom, depressions, or its edges, the sunshine duration is short, solar radiation is weak, groundwater is abundant, humidity conditions are better, and they mostly contain tall trees. On the slopes between the valley or depressions and the cliff top, the soil cover gradually decreases from the lower to the upper parts of the slopes, and the number of tree species also decreases in turn. The cliff tops consist of the peak cluster and cliffs. Since there is hardly any soil, the plants grow in the crevices of the rocks, which is common for arid vegetation. This area is home to several drought-tolerant plant species [29]. The study site is located south of the Tropic of Cancer and has a subtropical monsoon climate in the northern hemisphere with strong solar radiation, high evaporation, and rain and heat in the same season. The total rainfall during the study period was 1022 mm. Based on the available research results, we categorized the study period into rainy and dry seasons: April–September 2016 was the rainy season, and the remaining periods were denoted as dry seasons (January–March and October–December) [29].

A combination of sample plots and transects was used to investigate the vegetation structure within the François’ langur home range. A total of 12 sample plots and 10 sample transects were randomly set up, including 2 sample plots of 20 × 20 m, 2 sample plots of 15 × 20 m, 1 sample plot of 15 × 15 m, 1 sample plot of 10 × 20 m, 6 sample plots of 10 × 10 m, 4 sample transects of 40 × 5 m, 1 sample transect of 50 × 10 m, 2 sample transects of 80 × 5 m, 2 sample transects of 40 ×10 m, and 2 sample transects of 50 × 5 m, with a total survey area of 5625 m^2^. These plots and transects were set in the core area of the home range of the study troop, where a total of >70% of their daily activity took place (unpublished data) and were located in approximately 2% of their overall home range. All the plots and transects were randomly set up; however, we excluded those plots distributed on the cliffs due to their inaccessibility. Instead, we set the plots and transects based on the topographic conditions, consequently leading to the variety in the size and shape of those plots and transects. A total of 195 plant species belonging to 84 families were recorded in the sample survey of the François’ langur study area. Of these, the Euphorbiaceae, Rutaceae, and Pteridophyceae had the most abundant species, with 14, 11, and 11 species, respectively. The top 10 species by quantity were *Cipadessa cinerascens* (Meliaceae) (7.35%), *Desmos chinensis* (Annonaceae) (6.47%), *Litsea glutinosa* (Lauraceae) (5.80%), *Sterculiamonosperma* (Malvaceae) (2.80%), *Mallotus philippinensis* (Euphorbiaceae) (2.52%), *Croton euryphyllus* (Euphorbiaceae) (2.18%), *Catunaregam spinosa* (Rubiaceae) (1.77%), *Strophioblachia fimbricalyx* (Euphorbiaceae) (1.67%), *Cansjera rheedei* (Opiliaceae) (1.64%), and *Impatiens morsei* (Balsaminaceae) (1.49%).

The study langurs troop consisted of 12 individuals, including 1 adult male, 5 adult females, and 6 adolescent langurs. The langur group remained stable throughout the study period.

### 2.2. Data Collection and Analysis

We followed the langurs for a total of 55 days (including the days with incomplete data) from January to December 2016, observing them for 2–5 days per month. On each observation day, behavioral sampling began when the langurs were initially spotted. If the sleeping sites of the langurs from the previous day could be determined, behavioral sampling would begin at 6:00 A.M. the next day. These observations continued until the langurs entered their sleeping sites. In tracking the langurs, we sampled their behavior using the Instantaneous Scan Sampling method [30], with scan durations of 5 min and sampling intervals of 10 min. Specifically, within each 15-min sampling unit, the first 5 min was used for scanning the study troop, and the following 10 min was set for an interval between consecutive sampling units. To reduce the potential bias towards a given individual, all the langur members were scanned from left to right. We scanned as many different members as possible, but each individual was not sampled twice. The types of behaviors exhibited by the individuals seen at the sampling moment were recorded successively during the scan, and the behavior types were categorized as resting, moving, feeding, playing, grooming, and others [13,26,29]. While the sampled subjects were feeding, we recorded their feeding posture, forest strata height, and substrate sizes [13,26,31], with the related definitions detailed in Table 1. The sampling instance was terminated if the langurs did not appear within the observer’s field of view during the sampling time. In the end, we obtained a total of 2003 scans and recorded 12,585 individual langurs behaviors, including 4387 feeding records (Table 2).

For our data analysis, we took each feeding posture as a percentage of the total number of feeding postures recorded in that month as its frequency of utilization. The average of the percentages for each month was taken as the utilization frequency for the year. The same method was used to obtain data on the diameter and angle of the feeding substrate. The Mann–Whitney U test was used to compare the differences between two independent samples, and the Spearman correlation test was used to verify the correlation between the variables. There could be a potential risk of obtaining false significance values in the multiple correlations for variables. Thus, we reported the adjusted *p*-values using the FDR (False Discovery Rate) method for all correlations. All data analyses and tests were performed using R (Version 4.1.0) [32] with a significance level of 0.05.

## 3. Results

### 3.1. Dietary Composition and Their Seasonal Variation

Leaves are the main food type for François’ langurs, with young leaves accounting for 64.97% ± 19.08% of the food composition, mature leaves 11.88% ± 12.09%, fruits 12.96% ± 12.89%, flowers 4.16% ± 4.06%, and other food types, including stems, petioles, and other unknown parts of the tree, 6.03% ± 9.09%. There were significant seasonal variations in the food type composition of François’ langurs, mainly manifesting in the proportion of young leaves foraged by François’ langurs during the rainy season (74.94% ± 16.38%), which was significantly higher than that of the dry season (54.99% ± 17.15%) (z = −2.517, n = 12, *p* = 0.012). However, the feeding of mature leaves was inversed, with a significantly lower proportion of mature leaves foraged during the rainy season (4.21% ± 5.88%) than in the dry season (19.55 ± 12.09%) (z = −2.680, n = 12, *p* = 0.007). There were no significant seasonal differences in the utilization of flowers, fruits, and other parts by langurs (flowers: z = −0.568, n = 12, *p* = 0.570; fruits: z = −0.731, n = 10, *p* = 0.465; and other parts: z = −1.121, n = 12, *p* = 0.026).

### 3.2. Feeding Postures and Their Seasonal Variation

François’ langurs have four main postures when feeding: sitting, bipedal standing, quadrupedal standing, and suspending. Among them, the sitting posture was the most frequently adopted feeding posture by François’ langurs, followed by bipedal standing, accounting for an average of 85.99% ± 5.97% and 12.33% ± 6.08% of the total records, respectively. Suspending and quadrupedal standing postures were rare and only appeared in occasional feeding activities during peak resting time periods, accounting for 1.39% ± 1.59% of the total records (Table 2). There were significant seasonal variations in the feeding postures of François’ langurs (Figure 1), as evidenced by a significantly higher proportion of sitting feeding relative to bipedal standing feeding during the rainy season than during the dry season (z = −2.192, n = 12, *p* = 0.028), whereas François’ langurs adopted bipedal standing feeding with a significantly higher frequency during the dry season than during the rainy season (z = −2.842, n = 12, *p* = 0.004). Other feeding postures showed no significant seasonal variation due to their infrequent occurrence (z = −1.218, n = 12, *p* = 0.223) (Figure 1).

The feeding postures of François’ langurs were closely related to their food composition (Table 3). Correlation analyses showed that feeding posture was correlated with dietary composition, showing a positive correlation between the proportion of bipedal standing feeding and mature leaf consumption. However, the remaining feeding postures were not significantly correlated with food composition (Table 3).

### 3.3. Forest Strata Use by François’ Langurs during Feeding

François’ langurs preferred to feed in the lower and middle forest layers (Table 4 and Figure 2), accounting for 55.93% ± 16.50% and 33.63% ± 18.33% of the total number of records, respectively. Langurs were less likely to forage on the ground (rocks), accounting for only 6.79% ± 4.78% of the records. The frequency of langurs feeding in the upper part of the forest layer was the lowest at 3.65% ± 2.73%. When comparing forest strata heights during the dry and rainy seasons, we found that François’ langurs utilized the ground significantly more frequently during the dry season (9.56% ± 4.08%) than during the rainy season (4.01% ± 3.91%) (z = −2.355, n = 12, *p* = 0.019). Similarly, langurs utilized the lower forest layer more frequently during the dry season (65.58% ± 8.36%) than during the rainy season (46.28% ± 17.48%) (z = −2.192, n = 12, *p* = 0.028), whereas they utilized the middle forest layer less frequently during the dry season period (22.11% ± 8.22%) than during the rainy season (45.15% ± 18.78%) (z = −2.680, n = 12, *p* = 0.007). There were no seasonal differences in the utilization of the upper forest layer (2.75% ± 2.39% vs. 4.56% ± 2.96%, respectively, z = −1.218, n = 12, *p* = 0.223) (Figure 2).

The diameter of the substrates used by François’ langurs for feeding was dominated by small-to-medium-sized tree branches (including canes and shrubs) (Table 4 and Figure 3). Specifically, small branches accounted for 77.72% ± 10.55% of the feeding records, medium branches accounted for 14.49% ± 7.88%, and large branches accounted for 1.45% ± 2.75%. In addition, the ground was the substrate for 6.45% ± 5.46% of the langurs’ feeding activities. There was a seasonal difference in the size of the substrates used by the langurs for feeding, as evidenced by the more frequent ground utilization during the dry season (10.12% ± 4.98%) than during the rainy season (2.58% ± 2.55%) (z = −2.842, n = 12, *p* = 0.004). However, there was no significant seasonal variation at other sites (small: z = −0.568, n = 12, *p* = 0.570; medium: z = −0.406, n = 12, *p* = 0.685; and large: z = −0.661, n = 12, *p* = 0.508) (Figure 3).

The feeding habits of François’ langurs were found to be closely linked with the height of the forest strata they preferred to feed on. This relationship was observed in Table 5 and was supported by the negative correlation between the langurs’ feeding duration on young leaves and the frequency of using the lower forest layer. Additionally, there was a positive correlation between the frequency of utilizing the middle forest layer and the feeding duration on young leaves. However, there was no correlation between langurs’ frequency of the strata size use during feeding and the dietary composition (see Table 6).

## 4. Discussions

Food composition affects the selection of feeding postures in primates. In general, folivorous colobines mostly forage in a sitting position until they consume all the food within their reach [2,33]. When they need to change feeding sites, folivorous colobines generally stand up immediately and promptly move to the next feeding site, with no intermediate pauses [34]. However, when foraging for more dispersed foods such as fruits, primates need to move more frequently, and feeding postures need to facilitate their rapid movement, increasing the proportion of standing-position feedings [33]. In the present study, similar to most primates of the genus *Colobus*, François’ langurs rarely moved while feeding. When arriving at a feeding spot, they generally balanced their bodies to a comfortable position before feeding. After some minutes, they stopped feeding and moved to another location to start feeding again. François’ langurs mainly feed on leaves, which have a more even and stable distribution than other food types, such as insects [4,15,33], which may explain their use of the sitting position as their primary feeding posture. A previous study also confirmed that François’ langurs select sitting as their predominant feeding posture [13]. Actually, in other primates, particularly in colobines, sitting was the most frequently used stationary posture (see [13] for more details). Other primates also show similar trends [18].

Seated feeding in François’ langurs accounted for a significantly higher proportion of total behavioral records during the rainy season than during the dry season, whereas the frequency of bipedal standing feeding was significantly higher during the dry season than during the rainy season. This variation may be related to the spatial distribution of food resources. François’ langurs in this study were almost exclusively folivorous, preferring to eat young leaves and fruits while using mature leaves as fallback food. Unlike leaves, fruits tend to occur on smaller twigs located near the periphery of tree crowns [33,35]. Primates can cover adequate feeding areas by sitting while they feed on young leaves. During the rainy season, François’ langurs consumed more young leaves, which were more evenly and abundantly distributed in the tree crowns, probably allowing the langurs to harvest leaves while sitting. However, in the dry season, the study group exclusively used the mature leaves as fallback foods. The high-quality mature leaves were limited and gradually declined in availability by harvesting duration, causing the langurs to use bipedal standing to reach the leaves, now borne on less accessible twigs [13]. Actually, there were positive correlations between the bipedal standing frequency and the mature leaf consumption. A similar posture is also used by sympatric macaques [36], and the similar seasonal difference in feeding postures of François’ langurs has been linked to the variations in the food availability in the limestone forest and the dietary composition of these langurs [13]. The frequency of bipedal standing or forelimb suspending used by Assamese macaques was higher than in rhesus macaques because the former depended on the young parts of karst-endemic bamboo more heavily than the latter, and they always stood bipedally on the substrate and grasped the top stem where the young leaves grew [36]. Using bipedal standing by François’ langurs for feeding allows them to maximize their reach for foraging in the woods, which is less tiring than moving to another feeding site.

Primate body weight is one of the main factors limiting their forest strata height. In six primate species from the Tai Forest in the Ivory Coast of West Africa, the species with lower weights utilized smaller-diameter branches as their substrates more frequently than those with higher weights during feeding, which may be related to their body weight [33]. The preferred foods of all six of the aforementioned primate species are located in the canopy layer, especially in the outer layer. They tend to get as close as possible to the outer layers of the canopy while maintaining bodily balance, so weight becomes a limiting factor. François’ langurs have a body size close to that of other colobines and mainly feed on leaves, with a preference for young leaves [27,37]. Since young leaves are mainly found in the outer layers of the canopy, François’ langurs approach the outer layers of the canopy as much as possible to obtain young leaves while maintaining their balance. Most of the substrates used by François’ langurs are small, likely for the same reason.

The temporal and spatial distribution of food has implications for substrate selection during primate feeding. Food resources are not uniformly distributed in time and space but show cyclical variations [4]. Many primates have adapted their feeding behavior in response to this volatility in food availability. Dagosto [38] suggested that these changes in resource utilization will alter the selective usage of microhabitats by primates (use of different forest strata, etc.) or change their feeding patterns. Specifically, the effect of the spatial distribution of food on primate feeding substrates is manifested in the choice of forest strata height or feeding forest layer. For example, of the six primate species located in the Tai Forest in the Ivory Coast of West Africa, the three larger-bodied primates preferred to feed on smaller branches or thin branches (and were less likely to feed on thicker branches), and the three smaller-bodied species tended to feed more often on thin branches, mainly due to the distribution of food in the canopy [33]. The feeding activities of François’ langurs mainly occurred in the middle/lower canopy of valleys or foothill plains, with a forest strata height of approximate 5 m. This may also be related to the spatial distribution of food resources in the environment. Young leaves are the preferred food of François’ langurs, and similar to the six primary species of the Tai Forest, François’ langurs approach the outer canopy as much as possible to obtain young leaves while maintaining their body balance [13]. Seasonal changes in resources have caused corresponding changes in the feeding patterns of François’ langurs, including increased feeding duration during periods of food scarcity [39] and feeding on other low-quality foods [40]. The present study further confirmed that seasonal changes in food and other resources had a significant effect on the selective use of microhabitats by François’ langurs, mainly in the form of lower forest strata heights and greater use of the ground during the dry season. This change in microhabitat may be due to the change in the food composition of François’ langurs. During the dry season, the proportion of consumed mature leaves increased (this study; [41]), and feeding at forest strata heights below 2 m mostly occurred when the François’ langurs were searching for leaves from higher-quality vines in foothill plains [41]. For example, the kudzu (*Pueraria montana var. lobata*) vine mainly covers shrubs or small trees and is the primary species consumed by François’ langurs during the dry season (this study; [41]). The consumption of large numbers of mature leaves from this plant reduces forest strata heights, making it easier for François’ langurs to feed on young leaves during the rainy season. Since young leaves are mainly derived from arboreal species, the increased feeding of young leaves by François’ langurs led to increased forest strata height. Correlation analyses also showed that the proportion of young leaves in the langurs’ diet was significantly and positively correlated with the frequency of the langurs’ utilization of the middle forest layer.

Primate posture is affected by temperature [13,24,42,43]. In this study, there were significant seasonal differences in ground utilization by François’ langurs while feeding. At our study site, the ambient temperature during the dry season was lower than during the rainy season. Additionally, the limestone forest is characterized by a carbonatite matrix, resulting in a lower bare rock surface temperature during the dry season and a higher temperature than under tree crowns during the rainy season [44]. During the cold months of the dry season (especially December–February), the langurs utilize hilltops and cliff faces more frequently, mainly due to the tendency of langurs to bask in these areas to reduce their energy expenditure. Therefore, langurs forage for mature leaves on mountaintop cliffs to reduce their energy expenditure and conserve their energy by basking in the sun. Actually, these langurs used the ground (almost in the cliffs and hilltops) as substrates more frequently during the dry season than during the rainy season, likely due to their energy-conserving strategy. A similar strategy was adopted by the white-headed langurs (*Trachypithecus leucocephalus*) in the karst mountains [26].

In summary, the François’ langurs show four predominant feeding postures, of which sitting and bipedal standing were the most frequently used during feeding. These langurs tend to use the lower and middle forest layers as their feeding zones. Moreover, their feeding posture and substrate use seasonally vary according to their dietary composition. Our results demonstrate that the spatial distribution of foods in the limestone forest has an important effect on the feeding posture of the François’ langurs and their forest layer utilization, highlighting the importance of understanding the effects of ecological factors on the adaptation of the langurs to the unique limestone habitat.

## Figures and Tables

**Figure 1 animals-14-00565-f001:**
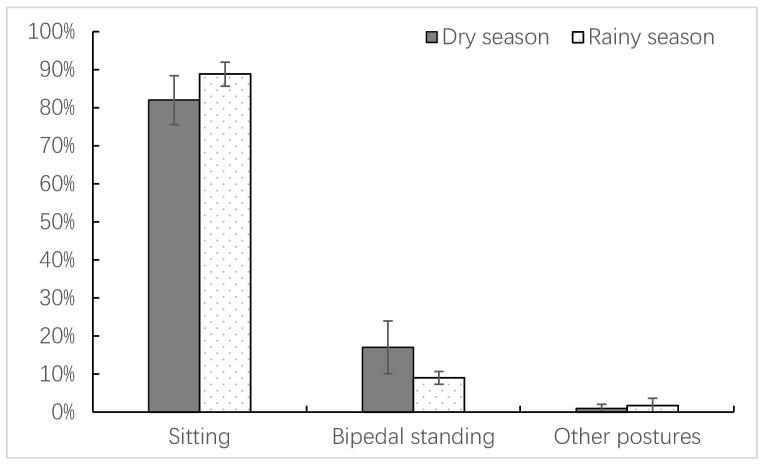
Seasonal variations in the feeding postures of the study langurs.

**Figure 2 animals-14-00565-f002:**
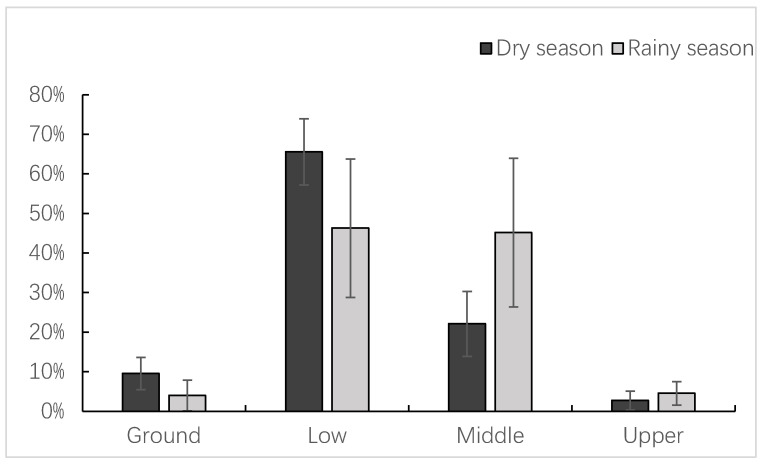
Seasonal variations in the forest strata heights during the feeding of the study langurs.

**Figure 3 animals-14-00565-f003:**
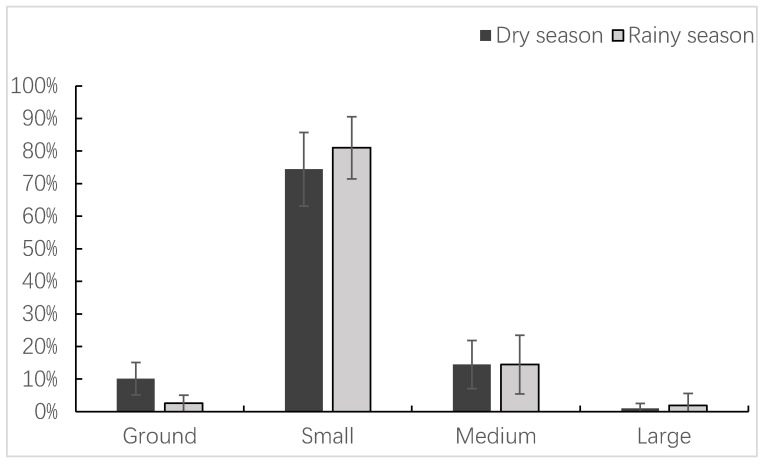
Seasonal variations in substrate sizes during the feeding of the study langurs.

**Table 1 animals-14-00565-t001:** Definitions of feeding behavior, forest strata height, and substrate size in this study (in accordance with Chen et al. [13], Hunt et al. [31], and Zheng et al. [26]).

Term	Definition
Activities	
Resting	The unaltered position of the langur individual
Feeding	The foraging, picking, ingestion, and chewing of food
Moving	Including quadrupedal walking, leaping, vertical climbing, and quadrupedal running
Social grooming	Mutual grooming behavior among the individuals
Postural modes	
Sitting	An orthograde posture in which the ischia bears a substantial portion of the body weight
Bipedal standing	Standing on the hind limbs with no significant support from any other body part.
Quadrupedal standing	Four-limbed standing on horizontal or subhorizontal supports; the elbow and knee are relatively extended and the trunk is near horizontal
Suspending	A part of the body hangs on the support, including the hind limb.
Forest strata height	
Ground	On the bare rock
Low	Height of ≤5 m of strata
Middle	Height of >5 m and ≤10 m
Upper	Height of >10 m
Substrate size	
Small	Substrate diameter of ≤2 cm
Medium	Substrate diameter of >2 cm and ≤5 cm
Large	Substrate diameter of >5 cm

**Table 2 animals-14-00565-t002:** Monthly feeding postures of François’ langurs.

Month	Total Sampling Days	Full Observation Days	Scan Records	Scan Individual	Feeding Records	Feeding Postures (%)		
						Sitting	Bipedal Standing	Other Postures
1	6	5	194	1029	342	85.13	14.29	0.58
2	5	4	191	1345	414	81.82	15.15	3.03
3	5	5	202	1415	496	85.91	10.15	0.38
4	5	5	222	1584	521	91.11	7.27	1.62
5	6	5	221	1284	404	91.25	8.00	0.75
6	4	4	156	804	337	92.04	6.78	1.18
7	4	4	144	813	320	89.09	10.30	0.61
8	5	5	168	1022	342	89.12	9.41	1.47
9	2	2	65	370	138	83.33	10.87	5.80
10	4	4	139	1054	358	70.13	29.44	0.43
11	5	4	166	989	393	84.93	14.87	0.20
12	4	3	135	876	322	87.99	11.36	0.65
Mean	4.58	4.17	166.92	1048.75	365.58	85.99	12.33	1.39
SD	1.08	0.94	44.04	326.76	97.00	5.97	6.08	1.59

**Table 3 animals-14-00565-t003:** Correlations among the feeding postures and dietary composition of the study langurs.

Posture	Statistics (n = 12)	Dietary Composition				
		Young Leaves	Mature Leaves	Flowers	Fruits	Others
Sitting	r	0.510	−0.580	−0.126	0.133	−0.399
	Adjusted *p*	0.225	0.225	0.697	0.697	0.332
Bipedal standing	r	−0.636	0.713	0.175	−0.168	0.510
	Adjusted *p*	0.065	0.045	0.602	0.602	0.150
Other postures	r	0.196	−0.147	−0.217	0.357	−0.007
	Adjusted *p*	0.811	0.811	0.811	0.811	0.983

**Table 4 animals-14-00565-t004:** Forest strata height and substrate size of François’ langurs during feeding.

Month	Forest Strata Heights (%)				Substrate Sizes (%)			
	Ground	Low	Middle	Upper	Ground	Small	Medium	Large
1	16.33	60.06	20.70	2.92	19.24	53.06	23.62	4.08
2	9.09	66.67	18.18	6.06	9.09	75.76	15.15	0
3	3.81	53.43	37.94	4.82	4.31	71.95	23.10	0.63
4	2.58	72.54	20.84	4.04	3.39	67.21	29.24	0.16
5	2.75	51.50	40.25	5.50	1.25	82.50	15.25	1.00
6	1.18	35.99	53.69	9.14	1.77	84.07	13.57	0.59
7	7.58	56.97	30.00	5.45	7.27	73.94	18.48	0.30
8	10.00	36.76	52.94	0.29	1.76	83.53	5.29	9.41
9	0	23.91	73.19	2.90	0	94.93	5.07	0
10	7.79	76.62	15.15	0.43	7.79	83.55	8.66	0
11	9.98	72.41	17.61	0	9.59	79.65	9.59	1.17
12	10.39	64.29	23.05	2.27	10.71	82.47	6.82	0
Mean	6.79	55.93	33.63	3.65	6.35	77.72	14.49	1.45
SD	4.78	16.50	18.33	2.73	5.46	10.55	7.88	2.75

**Table 5 animals-14-00565-t005:** Correlations among forest strata heights during feeding and the dietary compositions of the study langurs.

Forest Strata Heights	Statistics (n = 12)	Dietary Composition				
		Young Leaves	Mature Leaves	Flowers	Fruits	Others
Ground	r	−0.448	0.545	0.175	−0.245	0.392
	Adjusted *p*	0.347	0.335	0.587	0.554	0.347
Low	r	−0.741	0.392	0.070	0.007	0.406
	Adjusted *p*	0.030	0.347	0.983	0.983	0.347
Middle	r	0.713	−0.517	−0.077	0.077	−0.406
	Adjusted *p*	0.045	0.213	0.812	0.812	0.318
Upper	r	0.434	−0.531	0.056	0.287	−0.580
	Adjusted *p*	0.265	0.188	0.863	0.458	0.188

**Table 6 animals-14-00565-t006:** Correlations among strata sizes during feeding and the dietary compositions of the study langurs.

Strata Sizes	Statistics (n = 12)	Dietary Composition				
		Young Leaves	Mature Leaves	Flowers	Fruits	Others
Ground	r	−0.650	0.524	0.084	−0.049	0.329
	Adjusted *p*	0.110	0.200	0.880	0.880	0.495
Small	r	0.252	0.105	0.133	−0.462	0.455
	Adjusted *p*	0.717	0.746	0.746	0.345	0.345
Medium	r	−0.014	−0.329	−0.224	0.469	−0.678
	Adjusted *p*	0.966	0.495	0.605	0.310	0.075
Large	r	0.413	−0.142	−0.185	−0.164	−0.317
	Adjusted *p*	0.659	0.659	0.659	0.659	0.659

## Data Availability

The data presented in this study are available on request from the corresponding author. The data are not publicly available due to the project administration requirements.

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
