# Peer review of "Feeding Postures and Substrate Use of François’ Langurs (Trachypithecus francoisi) in the Limestone Forest of Southwest China"

_animals, 2024, doi:10.3390/ani14040565_

Round 1

Reviewer 1 Report

Comments and Suggestions for Authors

This is an interesting paper that addresses an important topic: the feeding Postures and Substrate Use of François’ langurs as aptly noted in the title of the manuscript.  I have identified a number of recommended edits on the attached document.  These include:

Line 17: delete 'method'

Line105:  you should provide citations to the previous studies which are noted. 

Line 109:  you note there have been damage and fragmentation to the habitat due to "historical reasons" -- you should state what these historical reasons are. 

Line 122: replace 'plans' with 'plains' 

Lines 136:  I suggest using 'plots' instead of 'squares' and 'transects' instead of 'strips' to conform with standard English and ecology terminology 

Lines 137-42 indicates that a variety of plot and transect sizes were used, you should briefly elaborate why this is the case.  I presume it relates to the complete topography of the research location but this should be clarified.  Also, on line 141 you note '40 x 10 m' but do not identify how many transects were of this size.

 Lines 142-43: delete 'The survey results showed that a' and insert 'A' to improve the English

 Line 152: replace 'colony' with 'troop' to conform to standard primatology terminology

Lines 159 and 161: replace 'nocturnal habitats' with 'sleeping sites' to conform to standard primatology terminology

Line 193: insert 'mature' before 'leaves'

Line 245: reword 'the substrate was the ground' to 'the ground was the substrate' to improve the English.

Lines 267, 269, and 313: preplace 'colobuses' with 'colobines' to improve English and to conform with standard primatology terminology

Lines 270-72: in this paragraph you are reporting on folivorous colobine behavior so using this example of a cercopithecine is not particularly pertinent -- I recommend that this sentence be deleted.

Line 276: 'Colobus' needs to be italicized

Line 400: delete 'New Yorks' and add 3rd Edition. 

Line 437-38:  You need to note that that Morbeck, Preuschoft, and Gomberg are editors of this volume, not authors.

 Line 448:  I believe a name is missing at the beginning of this citation.

Comments on the Quality of English Language

As a native English speaker I found the English in this manuscript to be acceptable for the most part.  I do note a number of needed edits above but I rarely had difficultly in understanding wha the authors are presenting.

Author Response

Dear reviewer,

Thank you for your comments. Current manuscript has been carefully revised according to the comments from editors and reviewers. Please see the attached files for details.

 Thank you in advance.

Sincerely,

Authors

Reviewer 2 Report

Comments and Suggestions for Authors

This is a meritorious study on the positional behavior and dietary ecology of Francois’s langurs in Guangxi, China. Major revisions to the methods section is needed to improve clarity and facilitate future study via replication and/or reproducibility, including but not limited to (1) elaboration, clarification, and correction of methods for sampling vegetation, (2) clarification and elaboration of scan sampling methodology, and (3) revision of statistical procedures for comparing proportions and adjusting for multiple comparisons. I didn’t carefully go through the statistical results because there were major problems in the stats that may impact significance testing. Minor revisions are needed to the introduction and discussion, and these improvements should be very easy to complete.  

Introduction

Lines 104-106: Cite the specific publications here on Françoise langur postural behavior in the Nongang National Nature Reserve. Also, in the discussion, compare and contrast your results to those of this earlier work.

Methods

Lines 136-142: Please elaborate on the randomization method used for placing sampling plots. What randomization technique was used? Also, provide a justification/rationale for the varying shapes (strips versus squares) and dimensions of vegetation plots to aid in study replication and repeatability in the future. There are several errors in the list of sample sizes for vegetation strips. 80 x 5 m strips and 50 x 5 m strips are each reported twice on this list. The number of 40 x 10 m strips is not provided. After these corrections are made, verify that the sampling area is accurate.

Lines 142-151: These sentences report the results of the vegetation survey, so they need to be removed from the methods section and moved to the results section.

Lines 176-178 (also all t-test and correlation tests in the results section): The student’s t-test is not appropriate for comparing proportions. Use the z-test instead as long as the scan samples are justifiably independent. Also, I didn’t see any adjustment or correction for the multiple comparisons performed in this analysis (comparison of proportions, correlation tests), so there is a risk of getting significant results from chance alone. There needs to be a posthoc correction added to the statistical analysis.

Line 157: There is a mismatch between the 3-5 days per month and Table 2, which shows that 2 is the minimum number of sampling days per month. Please adjust as appropriate.

Lines 162-163: Please elaborate on the scan sampling procedure concerning the 5-minute duration. Scans are usually quite short (e.g., 10 seconds or so) to take a snapshot of each individual’s behavior at a set point in time. Does the 5-minute duration mean that the team allotted up to five minutes to locate each individual in view and record their behavior one time during that 5-minute period? Then, the procedure was repeated 10 minutes later? Or was another protocol used? As written, there are too many unknowns to be able to clearly comprehend the sampling protocol. Is each individual represented only once per scan? Are you considering each scan to be independent of another scan? Could there be some bias in your scans due to visibility challenges of individual engaged in certain behavioral states (e.g., resting, feeding, moving) and how might this bias affect the statistical tests or interpretation of results?

Table 1: Ethograms of general behavior states, such as resting and feeding, tend to focus on the way that behavior is manifested by the organism, which may be defined at the individual or group level. Defining at the individual level is normative (except for social behaviors, such as grooming) because individuals within a group sometimes (but not always) engage in behaviors that are not identical to other group members, even though routinely many individuals are engaging in the same behavior at the same time. Defining at the individual level is more appropriate than the group level for the kinds of analyses in this study, unless it was assumed that all individuals were engaging in the same behavior at each scan. (And if this assumption was made, than it is imperative to elaborate on this detail in the methods and provide a clear justification for why this was done!). That is to say, many individuals are not a superorganism (although I suppose that is debatable). To standardize with the literature, assuming that each individual’s behavior was independently recorded at each sampling interval, then I suggest making minor changes to the appropriate definitions, such as “Feeding: The foraging, picking, ingesting, and chewing of food.” Please clarify if the definition of suspending as it relates to the tail. How does the Francois’s langur, that does not have a prehensile tail, suspend from it? Did you mean forelimb suspend instead or tail suspend? Please clarify.

 Results

See aforementioned concerns about the t-test and lack of posthoc corrections for the t-test and correlations.

Table 4: Please report the units (%?).

 Discussion

Contextualize these results with the other study(ies) on positional behavior in Francois’s langurs.

Lines 364-367: Please elaborate about how the results of this study lead to the conclusion that langurs forage for mature leaves on cliffs to conserve energy.

Author Response

(The authors gave the same response as above.)

Round 2

Reviewer 2 Report

Comments and Suggestions for Authors

The revised manuscript is much improved. I recommend that it be accepted for publication.

Comments on the Quality of English Language

Minor copy-edits to fix grammar errors are needed.